# Adiponectin Deficiency Alters Placenta Function but Does Not Affect Fetal Growth in Mice

**DOI:** 10.3390/ijms23094939

**Published:** 2022-04-29

**Authors:** Man Mohan Shrestha, Sanne Wermelin, Elisabet Stener-Victorin, Ingrid Wernstedt Asterholm, Anna Benrick

**Affiliations:** 1Unit for Metabolic Physiology, Department of Physiology, Institute of Neuroscience and Physiology, The Sahlgrenska Academy at the University of Gothenburg, 40530 Gothenburg, Sweden; man.shrestha@gu.se (M.M.S.); sanne.wermelin@outlook.com (S.W.); iwa@neuro.gu.se (I.W.A.); 2Department of Physiology and Pharmacology, Karolinska Institute, 17177 Stockholm, Sweden; elisabet.stener-victorin@ki.se; 3School of Health Sciences, University of Skövde, 54128 Skövde, Sweden

**Keywords:** adiponectin, placenta, fetal growth, triglycerides

## Abstract

Adiponectin administration to pregnant mice decreases nutrient transport and fetal growth. An adiponectin deficiency, on the other hand, as seen in obese women during pregnancy, alters fetal growth; however, the mechanism is unclear. To determine the role of adiponectin on placenta function and fetal growth, we used adiponectin knockout, adiponectin heterozygote that displays reduced adiponectin levels, and wild-type mice on a control diet or high fat/high sucrose (HF/HS) diet. Triglycerides (TGs) in the serum, liver, and placenta were measured using colorimetric assays. Gene expression was measured using quantitative RT-PCR. Adiponectin levels did not affect fetal weight, but it reduced adiponectin levels, increased fetal serum and placenta TG content. Wildtype dams on a HF/HS diet protected the fetuses from fatty acid overload as judged by increased liver TGs in dams and normal serum and liver TG levels in fetuses, while low adiponectin was associated with increased fetal liver TGs. Low maternal adiponectin increased the expression of genes involved in fatty acid transport; *Lpl* and *Cd36* in the placenta. Adiponectin deficiency does not affect fetal growth but induces placental dysfunction and increases fetal TG load, which is enhanced with obesity. This could lead to imprinting effects on the fetus and the development of metabolic dysfunction in the offspring.

## 1. Introduction

Pregnancy leads to substantial metabolic changes. Maternal fat stores increase during early gestation while late pregnancy is characterized by increased insulin resistance leading to increases in maternal glucose and free fatty acid concentrations [1,2], allowing for greater substrate availability for the growing fetus. Maternal obesity is the most common metabolic disturbance in pregnancy and a risk factor for negative health outcomes in both the mother and the fetus. The obese mother has an increased risk of pregnancy complications such as gestational diabetes, hypertension, pre-eclampsia, and cesarean section [2,3]. Maternal obesity also has a detrimental impact on the offspring’s health. One of the short-term adverse outcomes of maternal obesity is a fetal overgrowth, which is associated with metabolic disturbances later in life [2,4]. Fetal growth is directly determined by maternal nutrient availability and the placenta’s ability to transport these nutrients from maternal circulation to the fetus. The transplacental flux of nutrients is dependent on the availability and activity of nutrient-specific transporters [5,6,7,8]. Changes in the expression and activity of these transporters are associated with babies born small or large for their gestational age [9]. The placenta mediates the relationship between the maternal nutrient stores and fetal development by controlling nutrient transport and hormone production [2,5].

Adiponectin is an adipose-tissue-derived hormone that plays an important role in glucose homeostasis by increasing whole-body insulin sensitivity. There is increasing evidence that maternal adiponectin levels affect placenta function and fetal growth. However, the opposite effect is seen in the placenta, where adiponectin decreases insulin sensitivity [10,11]. Adiponectin is believed to play an important role in fetal growth as it inhibits placental insulin signaling and transport of glucose and amino acids, resulting in less nutrient availability for the fetus [10,12]. Adiponectin administration to pregnant mice decreases fetal growth [10,13,14]. In line with these mouse studies, human studies have found that lower maternal adiponectin levels are correlated with increased birth weight [15,16]. Adiponectin’s circulating levels are unlike other adipokines negatively correlated with body mass index, and thus adiponectin levels are lower in obese individuals [17]. Adiponectin levels decrease with each trimester of pregnancy for both lean and obese women, but the delta decrease in adiponectin is larger in normal-weight mothers compared to obese mothers, indicating higher metabolic flexibility in healthy-weight mothers [18,19,20,21]. The relationship between obesity, adiponectin, and fetal growth is, however, more complex, since low maternal adiponectin, as seen in overweight and obese women during pregnancy, is associated with both increased [15,16] and decreased [22,23] birth weight. Other studies fail to detect an association between maternal adiponectin and fetal body weight [16,24,25,26]. Moreover, the fetal production of adiponectin detected in cord blood also has a metabolic effect with a positive association with birth weight and fat mass [27,28,29]. There is clearly an interplay between maternal and fetal adiponectin, as well as other hormones and diet that regulate fetal growth, but the mechanism is not fully elucidated. Moreover, the effect of adiponectin depletion on fetal growth, studied in adiponectin knockout (APN ko) mice is uncertain, with contradictory results [13,30,31]. These studies report either a negative or a positive association between a complete lack of maternal adiponectin and fetal body weight. Previous studies include wild-type (wt) and APN ko dams, while this study also includes APN heterozygote (het) dams with 65% lower adiponectin levels. We propose that APN het mice are phenotypically a more relevant model to study adiponectin deficiency than APN ko mice since a complete lack of adiponectin likely leads to compensatory effects [32].

In this study, we investigate the effect of lowered adiponectin levels and a complete lack of adiponectin on placenta function and fetal growth using an APN ko mouse model [33]. The study design includes APN ko, APN het that displays reduced adiponectin levels, and wt mice fed a control diet (CD) or high fat/high sucrose diet (HF/HS) to define the role of maternal adiponectin levels and obesity on fetal growth. An adiponectin deficiency did not affect fetal growth but induced placental dysfunction and increased fetal triglyceride (TG) load, which was enhanced by obesity. This could lead to imprinting effects on the fetus and the development of metabolic dysfunction in the offspring.

## 2. Results

### 2.1. Maternal Variables

Non-pregnant APN ko and APN het mice on CD normally do not show a phenotype compared to wildtypes but develop metabolic disturbances, including glucose intolerance, insulin resistance, and delayed clearance of free fatty acids in plasma, when put on a high-fat diet [33]. Therefore, these mice were challenged with diet-induced obesity before and during pregnancy (Figure 1).

There was no difference in body weight between genotypes at six weeks of age. After eight weeks on HF/HS diet, there was a main effect of the diet with higher body weight, increased fat mass, and higher fasting insulin and blood glucose levels in the HF/HS fed group (Appendix A). There was no difference between genotypes except for insulin, which was higher in APN het and APN ko dams vs. wildtypes on CD (Appendix A), indicating more pronounced insulin resistance in mice deficient in adiponectin. A similar metabolic adaptation was observed in response to pregnancy at gestational day (GD) 18.5 in both CD and HF/HS fed dams, with no difference in fasting insulin or blood glucose levels between diets or genotypes. However, insulin was higher in pregnant dams compared to before mating in wt (*p* = 0.02), APN het (*p* = 0.001), and APN ko (*p* = 0.02) regardless of diet (Table 1 and Appendix A). Insulin resistance and resultant hyperinsulinemia are characteristic of pregnancy to deliver enough nutrients to the growing fetus. The hyperinsulinemia was able to maintain normal blood glucose levels in the pregnant dams, with no difference between groups (Table 1). The weights of the subcutaneous fat, brown fat, heart, and liver were heavier in the HF/HS groups compared to the CD groups, with no difference in body weight between genotypes (Table 1). Moreover, the body weights at GD 18.5 did not differ between diets, indicating a decreased gestational weight gain in the obese HF/HS fed dams. APN het dams on NC had a lower body weight and subcutaneous fat mass compared to wt dams (Table 1). There was no difference in the weights of the ovaries, mesenteric fat, pancreas, kidneys, or skeletal tibialis muscles between the groups (data not shown).

Circulating levels of total adiponectin were about one-third in APN het mice compared to wt (Figure 2A). As expected, adiponectin was undetectable in APN ko mice (Figure 2A,B). The more biologically active high-molecular-weight (HMW) adiponectin was further decreased in APN het mice and was only about one-tenth of wildtype levels (Figure 2B). Hence, the total/HMW ratio was lower in APN het mice compared to wildtypes (Figure 2C).

### 2.2. Fetal Variables

A complete lack of or lowered adiponectin levels did not alter fetal weight (Figure 3A). Maternal obesity decreased the litter size and fetal weight in all groups (Figure 3A,B). The decreased body weight in fetuses from obese dams was further enhanced in APN ko fetuses from heterozygote dams compared to wildtypes (Figure 3A). There was no difference in placenta weight (Figure 3C). Serum adiponectin levels were measured on blood samples pooled from male and female fetuses from each litter. Fetal serum samples from APN het dams are thus a mix of blood from APN ko and APN het fetuses. There was a tendency towards lower adiponectin levels in fetuses carried by APN het dams due to the fact that serum was collected from blood samples pooled from both APN het and APN ko fetuses. Some samples were collected only from APN ko fetuses, for example, if all females in a litter were knockouts. Adiponectin was undetectable in these samples. The total adiponectin levels were about one-third in APN het and APN ko fetuses carried by APN het or APN ko dams compared to wt fetuses from wt dams (Figure 3D). Fetal adiponectin levels were slightly lower in fetuses from diet-induced obese dams (Figure 3D).

### 2.3. Triglycerides

Pregnant dams challenged with diet-induced obesity had lower fasting serum TG levels. The higher insulin levels in HF/HS fed dams likely suppress the TG levels in the fasting state, while one would assume that the TG levels are higher in fed dams. The HF/HS diet led to a 2-fold increase in liver TG content compared to mice on CD (Figure 4A,B). No difference could be seen in the serum or liver TG levels with respect to genotype. Fetuses from dams on the HF/HS diet had slightly lower serum TG levels, just like the dams (Figure 4C). Fetuses from the APN het dams on the CD had higher serum TG levels compared to wt controls. There was no difference in TG levels between genotypes in fetuses from obese dams (Figure 4C). Wt dams on the HF/HS diet protected their fetuses from fatty acid overload as judged by normal serum and fetal liver TG levels in wt fetuses (Figure 4C,D). In contrast, adiponectin deficient (APN het) and APN ko dams on the HF/HS diet carried fetuses with an increased liver TG content compared to fetuses from obese wildtypes (Figure 4D).

### 2.4. Placenta

A two-way ANOVA was performed to explore the effects of the genotype and diet on the TG content in the placenta. There was a significant main effect for genotype (F_(2, 58)_ = 5.10, *p* = 0.018) but not diet. An increase in placenta TG concentration was seen in APN het dams on the CD and APN ko dams on the HF/HS diet compared to wt controls (Figure 5A). There was an interaction between diet and genotype (F_(2, 58)_ = 9.60, *p* = 0.0003). The glycogen content was similar between groups (Figure 5B).

The increased fetal serum and placenta TG levels in the APN het group on CD were associated with increased expressions of *Lpl* and *Cd36* in the placentas (Figure 5C,D). Placentas from APN het fetuses, regardless of the dam’s genotype, had increased expressions of *Lpl, Cd36*, and *Pparg (*Figure 5C–E). The expressions of *Lpl* and *Pparg* were not different in placentas from APN ko fetuses compared to wildtypes, and placentas from APN ko fetuses had lower expressions of *Lpl* compared to placentas from APN het littermates (Figure 5C,E). The difference between APN het and APN ko fetuses implies that not only maternal but also fetal adiponectin levels are involved in the regulation of placenta gene expression. None of the investigated genes were altered in the placentas from APN ko dams on the HF/HS diet compared to wt dams on the HF/HS diet, thus indicating that other genes are involved in the increased placenta TG content in this group. The HF/HS diet decreased the gene expression of several amino acid transporters; *Slc7a5, Slc38a4, Snat2* (Appendix A), fatty acid transporters; *Lpl, Cd36*, *Pparg (*Figure 5C–E)*,* and glucose transporter 3; *Slc2a3* (Appendix A). The decreased expression of nutrient transporters in the HF/HS fed mice could partly explain the decreased fetal weight. The placenta expression of *Ppargc1a, Vegf* and the VEGF receptor *Prokr1* (Appendix A) were also decreased in the HF/HS fed group, with only very minor effects of adiponectin deficiency.

## 3. Discussion

Adiponectin is present in three major multimeric forms, and HMW adiponectin is considered to be more biologically active as it has a longer half-life [17]. Low levels of HMW, and the ratio between the total and the HMW adiponectin, rather than the total amount of adiponectin is correlated to metabolic disorders [17]. Maternal adiponectin levels were less than half in APN het mice compared to wildtypes, with a further decrease in HMW adiponectin. Based on this finding, we propose that APN het mice present a more phenotypically relevant model to study adiponectin deficiency during pregnancy, than congenital APN ko mice. Moreover, a complete lack of adiponectin likely leads to compensatory effects as demonstrated in a recent study [32]. In general, the phenotype in APN het mice is not enhanced in APN ko mice [30].

There was a similar metabolic adaptation in response to pregnancy in normal weight and obese dams regardless of adiponectin levels, as judged by unaltered fasting glucose levels and a similar degree of hyperinsulinemia. This is in contrast to clinical data showing a strong association between low adiponectin levels and the development of gestational diabetes [34,35]. A lack of adiponectin has also been shown to impair glucose homeostasis in pregnant mice [13,36]. This difference between our and previous studies could be due to the APN ko mice that were generated by different labs, although they are all on a C56Bl/6 background. We and Qiao et al. [13,30,31] used APN ko mice generated by Scherer’s lab [33], while another study used mice generated by Scalia’s lab [36]. Moreover, the HF/HS diet did not alter adiponectin levels, which could be explained by the decreased gestational weight gain in the HF/HS fed dams, resulting in similar body weights at gestational day 18.5. Furthermore, maternal obesity decreased the litter size in all groups, and a lower number of fetuses decreased the dam’s body weight gain. If we subtract the sum of the fetuses’ weights from each dam’s body weight, there is a trend towards higher body weight in the HF/HS diet groups (F_(1, 39)_ = 33.31, *p* = 0.081). This is in line with the increased subcutaneous fat mass and ectopic fat accumulation in the liver and brown adipose tissue in the HF/HS fed dams.

Adiponectin produced by the dam does not pass the placental barrier, as adiponectin is too large (30-kDa protein) to theoretically pass the placenta. This is supported by detecting circulating adiponectin in adiponectin heterozygote dams but not in knockout fetuses, as shown in this study and by others [30]. Adiponectin produced by the fetus cannot pass the placenta either, since no adiponectin was detected in serum from APN ko dams carrying APN het fetuses. Similar findings in humans indicate that adiponectin in cord blood is derived from fetal and not from placental or maternal tissues [37]. Therefore, maternal and fetal adiponectin are not interchangeable and likely play different roles in fetal development. Maternal adiponectin acts on the placenta, not directly on the fetus, to alter nutrient transport. Fetal adiponectin, on the other hand, may have metabolic effects on the fetus, for example, to alter lipid metabolism in the liver. Fetal adiponectin may also affect the fetal part of the placenta as indicated by differences in placental gene expression between APN ko and APN het littermates. Whether the placenta is able to synthesize adiponectin is still in dispute. Adiponectin protein can be detected in crude placenta from humans and mice, but this likely reflects adiponectin levels in blood trapped within the intervillous blood spaces, since adiponectin cannot be detected in perfused placentas [38].

Few studies have investigated the association between the complete lack of maternal adiponectin levels and fetal body weight in mice, and those that have have found contradictory results from the same group. These studies report either a negative or a positive association between the complete lack of maternal adiponectin and fetal body weight [13,30,31]. We found no change in fetal body weight in dams with reduced levels or a complete lack of adiponectin. Maternal adiponectin has been suggested to reduce nutrient transport across the placenta by diminishing insulin/insulin-like growth factor 1 signaling in the placenta, resulting in reduced nutrients being available for the fetus and reduced fetal growth [11]. Clinical data might suggest that this signaling is particularly important when adiponectin levels are low, as in obese women, leading to increased nutrient transport and fetal growth [15,16]. However, a cohort of obese women indicates that low maternal adiponectin levels are more important for infant fat mass accumulation during the first months after birth, rather than birth weight [39]. Increased fat mass at GD 17.5, and during the first month after birth, is also evident in APN het offspring from APN ko dams [30]. Therefore, maternal adiponectin levels might also influence the offspring’s energy metabolism after birth through fetal programming. This is in line with improved metabolic health in wildtype offspring born from dams with elevated adiponectin levels [40]. We and others have shown that reduced levels or a lack of maternal adiponectin are associated with changes in the placenta and fetal liver function [30], which can have imprinting effects.

Low maternal adiponectin in the presence of fetal adiponectin enhances fat deposition in the offspring [15,16,30]. APN het fetuses from APN ko dams have more fat mass compared to APN ko littermates early in life [30]. However, one needs to bear in mind that fat mass in prenatal mice reflects mainly brown adipose tissue [30],since mice are born with very small amounts of subcutaneous fat and no gonadal fat [41]. This suggests that adiponectin produced by the fetus stimulates brown adipose tissue mass prenatally. Fetal adiponectin depletion on the other hand seems to act through a different mechanism, since APN ko fetuses from APN ko dams have less fat mass at term and during the first month of age [30].

Mouse fetuses start to produce adiponectin around embryonic day 15 [30,42], and fetal adiponectin levels affect liver gene expression and function [30]. Fetal liver function plays a key role in regulating fetal growth, since the umbilical vein enters the liver first before nutrients reach the systemic circulation. Diet-induced obesity in combination with low adiponectin levels affects fetal liver function as demonstrated by increased liver TG content in fetuses from APN het and ko dams on a HF/HS diet, indicating an increased lipid load and liver dysfunction. Wt fetuses, on the other hand, had normal liver TG levels. The fetal lipid load is likely an interplay between placenta function and fetal liver function, which is orchestrated by both maternal and fetal adiponectin.

Circulating glucose and lipids are elevated in obese women compared to normal-weight women, and this is believed to not only affect the placental uptake but also its metabolism [2]. Many of the investigated genes in the placenta were downregulated in response to the HF/HS diet. One of the key energy sources for the placenta is mitochondrial β-oxidation of fatty acids [2]. Obesity leads to an oversupply of circulating lipids, which is proposed to exceed the capacity of the β-oxidation, which results in elevated intracellular placental lipid levels, increased storage of lipids, and fatty acid transfer to the fetus [2]. Lipoprotein lipase acts at one of the initial steps in the transfer of free fatty acids across the placenta. It is involved in the hydrolysis of TGs present in chylomicrons and very-low-density lipoproteins to generate fatty acids for the fetus. Free fatty acids can be subsequently esterified or oxidized until they are transferred by facilitated diffusion to the fetus via fatty-acid binding proteins, e.g., CD36 [43]. *Lpl* and *Cd36* expression increased in placentas from APN het dams, and an increased transplacental uptake and transfer of fatty acids could explain the increased TG content in the placenta and the elevated serum TG levels in the fetuses. Moreover, fetal production of adiponectin can also affect placenta function. Fetal adiponectin may enhance the placental fatty acid transport by increasing the gene expression of lipoprotein lipase and peroxisome proliferator-activated receptor gamma (PPARγ), as seen in placentas from heterozygote fetuses from APN het or APN ko dams. PPARγ enhances fatty acid uptake and accumulation in mouse placentas [44]. These findings support the hypothesis that low maternal adiponectin levels increase nutrient transport to the fetus, in particular fatty acids. Moreover, it highlights the importance of maternal adiponectin and its interplay with fetal adiponectin to regulate placenta function.

## 4. Materials and Methods

### 4.1. Mouse Model

APN ko and APN het mice on a C57Bl/6 background, and wt littermate controls were used [33]. Wt and APN het/APN ko mice were genotyped as previously described [45]. Animals had ad libitum access to food and water and were housed under standard conditions (12-h light–dark cycle) at the Laboratory of Experimental Biomedicine, at the University of Gothenburg. The animals were fed either with a CD (7.42% fat, 17.49% protein and 75.09% carbohydrate; Special Diets Services, Scanbur, Sollentuna, Sweden) or a hypercaloric HF/HS diet (40% fat, 17% protein, and 43% carbohydrate (35% sucrose), 4.67 Kcal/g; D12079B Research Diets, Brogaarden, Horsholm, Denmark), in combination with 20% sucrose water (S9379, Sigma-Aldrich, Merck life science AB, Solna, Sweden) supplemented with Vitamin mix (10 g/4000 Kcal, V10001, Research Diets) and Mineral mix (35 g/4000 Kcal (S10001, Research Diets) [46]. The Ethics Committee of the University of Gothenburg, Sweden approved the study, which was performed in accordance with the legal requirements of the European Community (Decree 86/609/EEC).

### 4.2. Experimental Design

Six-week-old female mice were fed either with a CD or a HF/HS diet for eight weeks before mating. Vaginal smears were then performed to determine estrous cycle stage [47]. Female mice in proestrus and estrus were mated with male mice overnight according to the outline in Figure 1. The presence of a vaginal plug confirmed mating and pregnancy-associated weight gain was recorded. Pregnant mice were dissected on GD 18.5 after 2 h of fasting. Blood glucose was measured from the tail using OneTouch™ Ultra 2 m. Serum was collected from blood samples that were centrifuged at 10,000× *g* for 5 min (Microvette, Sarstedt, Helsingborg, Sweden). Heart, ovaries, inguinal fat, mesenteric fat, brown adipose tissue, pancreas, liver, and kidney were dissected and weighed. Fetus and placenta were separated from the umbilical cord and membranes and then weighed. Livers and placentas were dissected from each fetus. To determine the sex of the fetuses the distance between the anus and sex organs was observed. The distance in male fetuses is longer than in females. Blood samples from males and females were collected in separate tubes and centrifuged at 10,000× *g* for 5 min. All samples were snap-frozen in liquid nitrogen and stored at −80 °C. A small piece from the tail was taken from each fetus for genotyping [45].

### 4.3. Dual-Energy X-ray Absorptiometry

Body composition was determined in mice on the CD or the HF/HS diet before mating using Lunar PIXImus Mouse Densitometer (Wipro GE Healthcare, Madison, WI, USA). Mice were anesthetized during the procedure using isoflurane inhalation (2%; Isoba vet.), and bone, fat, and lean body mass were analyzed [48].

### 4.4. Serum Measurements

Fasted serum insulin (10-1247-01, Mercodia, Uppsala, Sweden) and total and HMW adiponectin (47-ADPMS-E01, Alpco, Salem, NH, USA) were measured in dams at GD 18.5 using ELISA. Male or female fetal serum was pooled from each dam and total adiponectin was measured with mouse DuoSets ELISA (DY1119, R&D Systems, Minneapolis, MN, USA). Serum TG levels were analyzed using a TR210 kit (Randox, Crumlin, UK), with the following changes to the manufacturer’s instructions: 200 µL of reagent R1 was added to each well containing 2 µL standard, or serum, in duplicates. A plate reader (SpectraMax I3x, Molecular Devices, San Jose, CA, USA) was used to measure the absorbance.

### 4.5. Liver and Placenta Triglyceride Content

Liver TGs were extracted from the dams and fetuses; 30–40 mg of the liver was washed in 0.5 mL ice-cold PBS for 5 min and then homogenized in 5% IGEPAL (CA-630, Sigma Aldrich) diluted in distilled water using a TissueLyser (Qiagen, Hilden, Germany). The samples were then heated to 80 °C for 5 min, then cooled down to room temperature, and heated again to 80 °C for 5 min. Then, they were centrifuged at 14,800× *g* for 2 min to remove insoluble material. The supernatant was transferred to a new tube and analyzed using the Randox TG kit (TR210, Cumlin, UK) as described above. Placenta TGs were extracted from 20–30 mg tissue homogenized in 800 µL of 2:1 chloroform: methanol (*v*/*v*) mixture using a TissueLyser (40 Hz for 5 min), and then left on a nutating mixer for 20 min at room temperature. The samples were centrifugated at 16,200× *g* for 2 min and then washed with 160 µL 0.9% NaCl. Samples were centrifuged at 400× *g* for 2 min and 200 µL of the lower phase was collected and left to evaporate. The samples were resuspended in 75 µL isopropanol (Sigma-Aldrich, Merck life science AB, Solna, Sweden) and analyzed using the Randox TG kit (TR210).

### 4.6. Placenta Glycogen Content

A total of 10–12 mg of placenta was homogenized (40 Hz for 3 min) in 100 µL ice-cold water using a TissueLyser (Qiagen, Hilden, Germany). The homogenates were incubated at 85 °C for 5 min and centrifuged at 16,200× *g* for 5 min. Placenta glycogen content was analyzed in the supernatant using a Glycogen Assay Kit (MAK016-1KT, Sigma-Aldrich, Merck life science AB, Solna, Sweden) according to the manufacturer’s instructions. The absorbance was analyzed using a SpectraMax I3x plate reader.

### 4.7. Quantitative Real-Time PCR

RNA isolation from the placentas was performed using RNeasy Mini Kit (74,104, Qiagen). Isolation was performed according to the manufacturer’s instructions (*n* = 8/group). cDNA was prepared using SuperScript II Reverse transcriptase (18-064-014, Invitrogen) with the addition of RNaseOUT™ Recombinant Ribonuclease Inhibitor (10-777-019, Invitrogen, Waltham, MA, USA) according to the manufacturer’s instructions. A real-time PCR was run on a QuantStudio7 Flex instrument (Applied Biosystems, Waltham, MA, USA) with the following conditions: 95 °C for 7 min (1 cycle), 95 °C for 15 s, 60 °C for 1 min, 95 °C for 15 s (40 cycles), followed by a melting curve (0.5 °C/s) from 60 °C to 95 °C to confirm one PCR-product, as products were detected using Fast SYBR Green Master Mix (4,385,612, Thermo Fisher Scientific, Waltham, MA, USA). Primer sequences are found in (Appendix A). The RefFinder algorithm was used to determine the most stable reference gene. *Actb* had the lowest inter- and intragroup variability compared to *Gapdh* and *Hprt*1. Gene expression was calculated using the 2^−∆∆Ct^ method.

### 4.8. Statistical Analysis

Data are presented as the mean ± SEM and were analyzed with SPSS (version 28.0.1.0) and Prism (version 9.3.1). A Kolmogorov-Smirnov test was performed, and the data were not normally distributed. The main effects of diet (CD or HF/HS) or genotype (wt, APN het, APN ko) were measured using s two-way ANOVA. A Brown–Forsythe ANOVA test with Dunnett’s multiple comparisons test was run to compare if there was a significant difference between the genotypes within diets. A Mann–Whitney U test was used to analyze the adiponectin levels in dams and the differences between APN het and APN ko fetuses from the same litter. *p*-values < 0.05 were considered statistically significant.

## 5. Conclusions

Adiponectin deficiency in pregnant mice does not affect fetal growth but induces placental dysfunction and increases fetal TG load, which is enhanced in obese dams. Both maternal and fetal adiponectin levels are involved in the regulation of placenta gene expression. Our findings highlight the interplay between maternal and fetal adiponectin in the regulation of placenta function. This may lead to imprinting effects on the offspring, leading to increased fat mass gain and impaired metabolic health. Future studies are needed to investigate the imprinting effects of low maternal adiponectin levels in adult offspring.

## Figures and Tables

**Figure 1 ijms-23-04939-f001:**
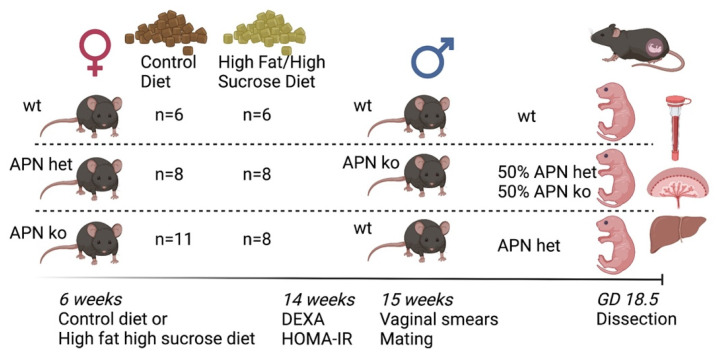
Study design. Adiponectin knockout (APN ko), adiponectin heterozygote (APN het), and wild-type (wt) female mice were fed either a control diet or a high fat/high sucrose diet for 8 weeks. Body composition by DEXA and fasting glucose and insulin levels were measured before mating. Vaginal smears were performed and mice in proestrus and estrus were mated with male mice overnight. Pregnant mice were dissected on gestational day (GD) 18.5. Serum, placentas, and livers were collected. The genotype is shown next to the females, males, and fetuses in each group. Created with BioRender.com (accessed on 7 April 2022).

**Figure 2 ijms-23-04939-f002:**
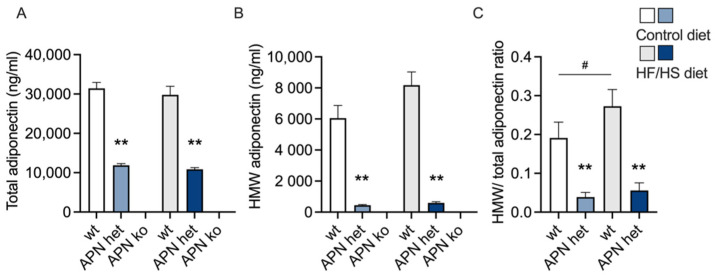
Total adiponectin (**A**), high-molecular-weight (HMW) adiponectin (**B**), and HMW/total adiponectin ratio (**C**) in adiponectin knockout (APN ko), adiponectin heterozygote (APN het), and wild-type (wt) dams on control or high fat/high sucrose (HF/HS) diet. The effect of genotype within diet was analyzed using the Brown-Forsythe ANOVA test with Dunnett’s multiple comparisons test. Data are presented as mean ± SEM. ** *p* < 0.01 vs. wt within the diet, ^#^
*p* < 0.05 vs. wt control.

**Figure 3 ijms-23-04939-f003:**
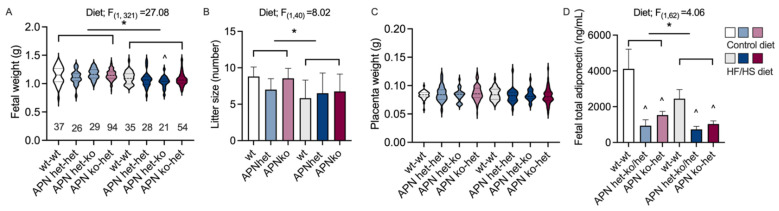
Fetal body weight (**A**), litter size (**B**), placenta weight (**C**), and fetal serum adiponectin levels (**D**) in fetuses from adiponectin knockout (APN ko, *n* = 11 Control, *n* = 8 HF/HS diet), adiponectin heterozygote (APN het, *n* = 8 Control, *n* = 8 HF/HS diet), and wild-type (wt, *n* = 6 Control, *n* = 6 HF/HS diet) dams on high fat/high sucrose (HF/HS) diet or control diet. APN het dams carried APN het and ko fetuses; when possible, these fetal groups (APN het-het and APN het-ko) were analyzed separately. The groups are labelled with the maternal genotype followed by the fetal genotype. The effect of diet-induced obesity and dam genotype (wt, APN het, and APN ko) was analyzed using two-way ANOVA, * *p* < 0.05. The effect of genotype within diet was analyzed using the Brown–Forsythe ANOVA test with Dunnett’s multiple comparisons test. Data are presented as mean ± SEM. The fetal *n*-number is given in (**A**). ^^^
*p* < 0.05 vs. wt within the diet.

**Figure 4 ijms-23-04939-f004:**
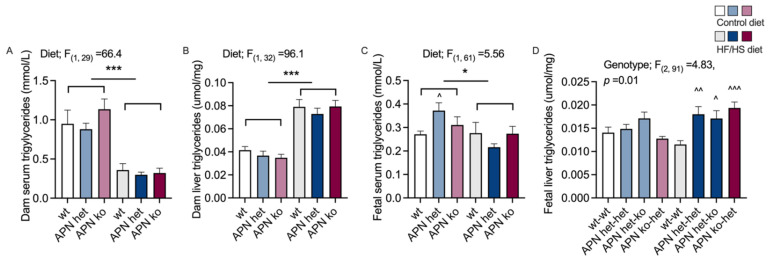
Serum triglycerides (**A**) and liver triglyceride content (**B**) in adiponectin knockout (APN ko), adiponectin heterozygote (APN het), and wild-type (wt) dams on control or high-fat/high-sucrose (HF/HS) diet. Serum triglycerides (**C**) and liver triglyceride content (**D**) in fetuses from wt, APN het and APN ko dams. APN het dams carried APN het and ko fetuses, when possible, these fetal groups (APN het-het and APN. het-ko) were analyzed separately, as in (**D**). The groups are labelled with the maternal genotype followed by the fetal genotype. The effect of diet-induced obesity and dam genotype (wt, APN het, and APN ko) was analyzed using two-way ANOVA, * *p* < 0.05, *** *p* < 0.001. The effect of genotype within diet was analyzed using the Brown–Forsythe ANOVA test with Dunnett’s multiple comparisons test. ^ *p* < 0.05, ^^ *p* < 0.01, ^^^ *p* < 0.001 vs. wt within diet. Data are presented as mean ± SEM.

**Figure 5 ijms-23-04939-f005:**
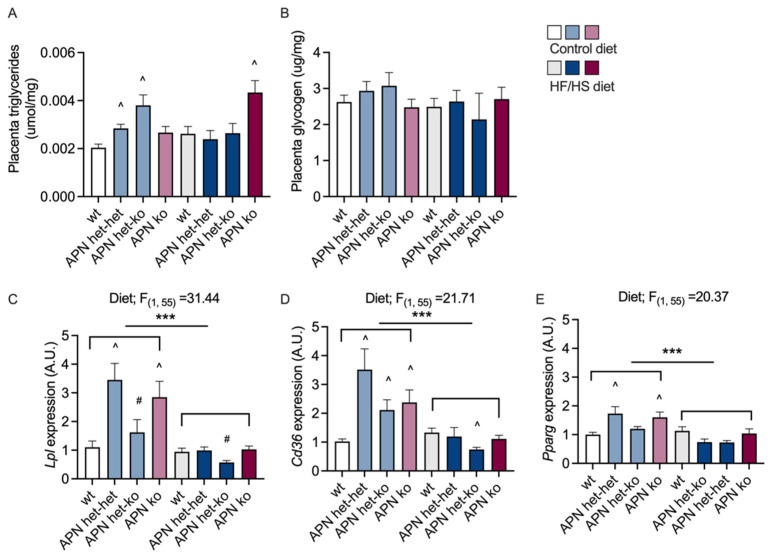
Placenta triglyceride (**A**) and glycogen (**B**) content, placenta gene expression of *Lpl* (**C**), *Cd36* (**D**), and *Pparg* (**E**) from adiponectin knockout (APN ko), adiponectin heterozygote (APN het), and wild-type (wt) dams on control or high-fat/high-sucrose (HF/HS) diet. The effect of diet-induced obesity and dam genotype (wt, APN het, and APN ko) was analyzed using two-way ANOVA, *** *p* < 0.001. The effect of genotype within diet was analyzed using the Brown–Forsythe ANOVA test with Dunnett’s multiple comparisons test. ^ *p* < 0.05 vs. wt within the diet. ^#^
*p* < 0.05 APN het-het vs. APN het-ko. Data are presented as mean ± SEM.

**Table 1 ijms-23-04939-t001:** Body weight, tissue weights, and serum measurements at GD18.5 in adiponectin knockout (APN ko), adiponectin heterozygote (APN het), and wild-type (wt) dams on control diet (CD) or high fat/high sucrose (HF/HS) diet.

Maternal Variables	Wt CD	APN Het CD	APN Ko CD	Wt HF/HS	APN Het HF/HS	APN Ko HF/HS	Two-Way ANOVA
	*n* = 6	*n* = 8	*n* = 11	*n* = 6	*n* = 8	*n* = 8	
Body weight (g)	40.9 ± 1.0	35.0 ± 0.9 **	41.2 ± 1.4	37.3 ± 1.9	38.3 ± 1.1	39.9 ± 0.7	
Blood glucose (mmol/L)	5.58 ± 0.16	6.66 ± 0.52	6.77 ± 0.42 *	6.26 ± 0.45	6.33 ± 0.20	5.49 ± 0.29	
Insulin (µg/L)	1.29 ± 0.33	1.16 ± 0.28	1.37 ± 0.36	0.78 ±0.25	1.98 ± 0.40	0.93 ± 0.29	
Heart (mg)	115 ± 3	113 ± 3	124 ± 4	131 ± 7	135 ± 4	150 ± 6	a; F_(1, 39)_ = 29.8, *p* < 0.001 b; F_(2, 39)_ = 5.57, *p* = 0.007
Subcutaneous inguinal fat (g)	0.62 ± 0.05	0.39 ± 0.03 **	0.52 ± 0.05	0.86 ± 0.18	0.72 ± 0.07	0.89 ± 0.10	a; F_(1, 38)_ = 19.9, *p* < 0.001
Mesenteric fat (g)	0.38 ± 0.03	0.29 ± 0.02	0.32 ± 0.04	0.35 ± 0.08	0.32 ± 0.03	0.39 ± 0.07	
Brown adipose tissue (mg)	62.6 ± 3.4	58.7 ± 4.6	72.3 ± 7.1	80.6 ± 8.7	69.4 ± 1.9	78.6 ± 6.6	a; F_(1, 39)_ = 5.73, *p* = 0.022
Liver (g)	1.57 ± 0.06	1.58 ± 0.07	1.74 ± 0.05	2.17 ± 0.12	2.32 ± 0.13	2.43 ± 0.142	a; F_(1, 39)_ = 65.9, *p* < 0.001

The effect of diet-induced obesity and dam genotype (wt, APN het, and APN ko) on tissue weights and serum measurements was analyzed using two-way ANOVA. a; main effect of diet, b; main effect of genotype. The effect of genotype within diet was analyzed using the Brown–Forsythe ANOVA test with Dunnett’s multiple comparisons test. * *p* < 0.05, ** *p* < 0.01 vs. wt within the diet. Data are presented as mean ± SEM.

## Data Availability

The data presented in this study are available on request from the corresponding author.

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
