# Peer review of "Adiponectin Deficiency Alters Placenta Function but Does Not Affect Fetal Growth in Mice"

_ijms, 2022, doi:10.3390/ijms23094939_

Round 1
Reviewer 1 Report
The study of Man Mohan Shrestha et al., titled “Adiponectin deficiency alters placenta function but does not affect fetal growth in mice”. The authors present a well-designed study, which convincingly showed that adiponectin deficiency in pregnant mice most likely affects placental function and fetal triglyceride load rather than fetal birth weight. This is an interesting finding that further highlights the distinction that needs to be made between maternal and fetal adiponectin, as it does not cross the placenta. Moreover, the distinct effect of HMW and overall adiponectin was well described. The main backdraw of the study was the mix of heterozygous and ko blood samples to determine total adiponectin content. If possible, adding a gene expression assay for each fetus (APN) might be desirable to further strengthen the conclusion.
Specific areas of improvement
-Major issues:
1) What was the reason for combining the blood samples from all fetuses? If it was low sample availability, what substitute measurement could have been used (e.g., gene expression)?
Adding a gene expression assay (APN) with data for each fetus is desirable to further strengthen the conclusion.
2) Please provide Western (or similar Blots) to substantiate the expression of APN in the knockout and heterozygous mice in different organs/blood.
-Minor issues:
1) Abbreviations. Please ensure that all abbreviations are elaborated at their first mention and that they are then used consistently throughout the manuscript.
2) You stated that the decrease of adiponectin is larger in normal weight mothers during pregnancy. Please clarify, whether this is also the case for HMW adiponectin. Along these lines, it would be expected to see a difference between adiponectin levels with different diets at baseline. How would you explain the similarity between the levels on different diets?
3) Please ensure to use the correct punctuation of “however” throughout the manuscript.
4) Please pay attention to the consistency of the abbreviations throughout the manuscript. Please choose either adiponectin, or APN and then stay consistent. The abstract may be an exception from this general rule. If in doubt, please refer to the formatting instructions of the journal.
5) Colloquial abbreviations, such as “don’t” should be avoided.
Line 75: When choosing to cite the studies relating to Adiponectin depletion in mouse models, please elaborate in more detail how the studies are contradictory in 1-2 sentences. Also, please elaborate what makes your study stand out amongst the previous one’s and what is the added knowledge that has been gained through your study.
Line 90: Please specify the metabolic disturbances.
Table 1: Please indicate all significant differences with asterices in Table 1. For example, Insulin has not been indicated to show a significant difference.
In Figure 3, you seem to indicate that the APN ko mice had only heterozygous offspring (APN ko-het). Is this correct?
In Figures 3 and 4, please indicate the pvalues using asteriscs above the brackets.
Line 206: This sentence may be overstating your findings. You have shown that fetal weight was decreased in ht-ko mice during high-fat diet. However, the effect of high-fat diet seems to span across multiple genotypes (i.e., Lpl expression is decreased from 3.5, 1.5, and 2.5 to 1, 0.5, and 1 across APN het-het, het-ko, and ko respecitively).
Line 253: “perfused placentas” may be a substitute term for “washed out of blood”
Line 278: Punctuation
Reviewer 2 Report
In this study, Shrestha and co-workers investigate the effect of lowered adiponectin levels and complete lack of adiponectin on placenta function and fetal growth using an adiponectin knockout mouse model. The work is generally well written and structured. The proposed topic is very interesting.
However, some clarifications are needed, in particular, some dietary controls are missing. Please, show data about weight gain, food intake, and water intake of the different groups during the 8-weeks diet. The clinical data show no effect in WT after HF/HS diet and chow diet; there are no significant differences in body weight, glucose, insulin, and so on. How do the authors explain these data?
Generally, there are no significant differences in the weight of different organs (heart, liver, etc.). Please, discuss this data. Are there differences in ectopic fat accumulation?
In normal pregnancy, most inflammatory cytokines in the maternal circulation increase across pregnancy, in part due to cytokine secretion by the placenta. How about cytokine secretion in these mice models?
Round 2
Reviewer 1 Report
Thank you for revising the manuscript according to the comments.
Reviewer 2 Report
The authors have included additional data to answer open questions and rephrased parts of the manuscript. The revised version addresses most of my previous criticisms satisfactorily. I have no additional comments.